# Expression Profiling Identified TRPM7 and HER2 as Potential Targets for the Combined Treatment of Cancer Cells

**DOI:** 10.3390/cells13211801

**Published:** 2024-10-31

**Authors:** Miyuki Egawa, Eva Schmücker, Christian Grimm, Thomas Gudermann, Vladimir Chubanov

**Affiliations:** 1Walther-Straub Institute of Pharmacology and Toxicology, Ludwig Maximilian University of Munich, 80336 Munich, Germany; miyuki.egawa@campus.lmu.de (M.E.); eva_schmuecker@gmx.de (E.S.); christian.grimm@med.uni-muenchen.de (C.G.); 2Immunology, Infection and Pandemic Research IIP, Fraunhofer Institute for Translational Medicine and Pharmacology ITMP, 80799 Munich, Germany; 3Comprehensive Pneumology Center, German Center for Lung Research (DZL), 81377 Munich, Germany

**Keywords:** TRP channels, HER2, ERBB2, NS8593, CP724714, zinc, breast cancer

## Abstract

TRPM7 is a divalent cation-permeable channel that is highly active in cancer cells. The pharmacological inhibitors of TRPM7 have been shown to suppress the proliferation of tumor cells, highlighting TRPM7 as a new anticancer drug target. However, the potential benefit of combining TRPM7 inhibitors with conventional anticancer therapies remains unexplored. Here, we used genome-wide transcriptome profiling of human leukemia HAP1 cells to examine cellular responses caused by the application of NS8593, the potent inhibitor of the TRPM7 channel, in comparison with two independent knockout mutations in the *TRPM7* gene introduced by the CRISPR/Cas9 approach. This analysis revealed that *TRPM7* regulates the expression levels of several transcripts, including *HER2* (*ERBB2*). Consequently, we examined the *TRPM7/HER2* axis in several non-hematopoietic cells to show that TRPM7 affects the expression of HER2 protein in a Zn^2+^-dependent fashion. Moreover, we found that co-administration of pharmacological inhibitors of HER2 and TRPM7 elicited a synergistic antiproliferative effect on HER2-overexpressing SKBR3 cells but not on HER2-deficient MDA-MB-231 breast cancer cells. Hence, our study proposes a new combinatorial strategy for treating HER2-positive breast cancer cells.

## 1. Introduction

The transient receptor potential cation channel, subfamily M, member 7 (TRPM7), is a bifunctional protein containing a membrane-spanning cation channel segment fused to a cytosolic protein kinase domain [1,2,3,4]. Electrophysiological experiments demonstrated that TRPM7 forms a constitutively active channel, which is highly permeable to divalent cations, including Zn^2+^, Mg^2+^, and Ca^2+^ [5,6,7,8,9]. Independent evidence supports the notion that the TRPM7 channel represents the main route for the cellular uptake of divalent cations, especially Mg^2+^ and Zn^2+^ [10,11,12,13,14]. In line with this assumption, the genetic disruption of TRPM7 causes cell cycle arrest [10,11,12]. In addition to the homeostatic control of Mg^2+^, the TRPM7 channel regulates multiple Zn^2+^- and Ca^2+^-regulated pathways pertinent to cancer progression [15,16,17,18,19,20].

The kinase domain of TRPM7 belongs to the atypical group serine/threonine-protein kinases entitled α-kinases (or alpha-protein kinases) [4,21,22,23]. TRPM7 and its close homologous protein TRPM6 are the only known ion channels covalently fused to kinase domains [2,4]. In addition, four other human proteins contain α-kinase domains, including eukaryotic elongation factor-2 kinase (eEF-2K) and alpha-protein kinases 1–3 (ALPK1–3) [21,22,23]. The identified phosphorylation substrates of TRPM7 kinase comprise a set of functionally heterogeneous proteins like TRPM6, annexin A1, myosin II, eEF-2K, tropomodulin, PLCγ2, STIM2, SMAD2, RhoA, and CREB [1,2,3,22,23].

Several drug-like molecules were identified as potent modulators of the TRPM7 channel [24,25,26,27,28]. Among them, NS8593, waixenicin A, and FTY720 represent the most comprehensively characterized inhibitors of the TRPM7 channel [25,29,30,31,32]. Recently, cryogenic electron microscopy (cryo-EM) structures of the TRPM7 channel in active open and inactive closed states were solved [4,33,34]. In addition, cryo-EM identified the binding site of NS8593 and delineated the molecular mechanism underlying its inhibitory impact on the TRPM7 channel [4,34]. Mechanistically, the pharmacological effects of waixenicin A and FTY720 on TRPM7 remain less understood.

NS8593 was extensively used to dissect cellular functions of the TRPM7 channel in different physiological and pathophysiological settings, including animal models of human diseases [25,35,36]. One of the most noticeable findings is that NS8593 showed inhibitory effects on the proliferation of cancer cells [10,20,35,37,38,39,40,41]. Additionally, many investigators used NS8593 or other TRPM7 inhibitors, often in combination with RNAi silencing of *TRPM7*, to demonstrate the crucial role of TRPM7 in signaling pathways and cellular processes linked to tumor progression [25,35,38,40,42,43,44,45], including breast cancer [38,46,47,48,49,50,51]. These findings correlate well with genetic, histological and bioinformatic analysis of human tissue samples, suggesting TRPM7 is aberrantly expressed in many cancer types and can serve as a prognosis marker of the disease progression and survival of the patients [52,53,54,55,56]. Consequently, TRPM7 has been proposed as a new anti-cancer drug target [56,57,58].

In the present study, we conducted transcriptome profiling of human leukemia HAP1 cells to characterize cellular responses after genetic inactivation and pharmacological inhibition of TRPM7. In follow-up experiments with HAP1 cells and different cancer cells, we found that TRPM7 regulates the expression of HER2 and that this regulatory mechanism can be exploited for combinatorial pharmacological treatment of HER2-expressing breast cancer cells.

## 2. Materials and Methods

### 2.1. Pharmacological Agents

NS8593 and CP724714 were acquired from Tocris, Bristol, UK. ZnCl_2_ and zinc pyrithione (#PHR1401) were purchased from Merck, Darmstadt, Germany.

### 2.2. HAP1 Cells

Parental (herein referred to as wild-type (WT) cells) and *TRPM7* knockout (*TRPM7* KO) human haploid leukemia (HAP1) cells were acquired from Horizon Discovery (Cambridge, UK). The genetic and functional characterization of clone 10940–04 (referred to as *TRPM7* KO-04) was described previously [12,39]. A CRISPR/Cas9 approach was used to introduce a 17 bp (GTGACCATTTTAATCAG) deletion in exon 4 of the human *TRPM7* gene, resulting in a frame-shift mutation [12,39]. The clone 10940–01 of HAP1 cells (referred to as *TRPM7* KO-01) was characterized in the present study. A 16 bp targeting sequence (AAAATGGTCACCCAAT) was selected to modify the exon 4 of *TRPM7* using the CRISPR/Cas9 technique. The *TRPM7* KO-01 clone was examined by RT-PCR approach using *hTRPM7*-Forward 5′-GGAGTCCGCCCCGTGAGG-3′ and *hTRPM7*-Reverse 5′-TGACTTCCGCCCCATACTTTCCAACAG-3′ primers and the following PCR settings: 95 °C 30′′, 63 °C 15′′, 72 °C 90′′. The obtained PCR products were isolated from the gel, purified and sequenced. The sequence of PCR product from WT cells matched to *TRPM7* mRNA. The PCR product from *TRPM7* KO-01 cells contained a sequence of exon 3, which was aberrantly spliced to an intronic sequence followed by a sequence from the distal part of exon 4 of *TRPM7* (Appendix A). In silico translation of the mutant cDNA revealed a frame-shift mutation in *TRPM7*.

### 2.3. Cell Cultures

HAP1 cells were cultured in IMDM supplemented with 10% fetal bovine serum (FBS), 100 U/mL penicillin and 100 µg/mL streptomycin mixture (all from Merck, Darmstadt, Germany). Unless indicated otherwise, *TRPM7* KO-01 and KO-04 HAP1 cells were supplemented with 10 mM MgCl_2_ included in the regular cell culture medium.

WT and *TRPM7* knockout (*TRPM7* KO) mouse trophoblast stem (TS) cells were derived from mouse blastocysts as described earlier [39]. TS cells were cultured in RPMI 1640 medium (Merck, Darmstadt, Germany) supplemented with 20% FBS (ES type, Life Technologies, Darmstadt, Germany), 1 mM sodium pyruvate (cell culture type, Merck, Darmstadt, Germany), 100 mM β-mercaptoethanol (Merck, Darmstadt, Germany), 50 µg/mL streptomycin (Merck, Darmstadt, Germany), 50 U/mL penicillin (Merck, Darmstadt, Germany), 1.0 mg/mL heparin (cell culture type, Merck, Darmstadt, Germany), 25 ng/mL human recombinant FGF4, 5 ng/mL human recombinant TGF-β1, and 10 ng/mL recombinant activin A (all from R&D systems, Minneapolis, MN, USA). Cells were maintained in a humidified cell culture incubator (Heraeus, Thermo Fisher Scientific, Waltham, MA, USA) at 37 °C and 5% CO_2_.

*TRPM7* KO HEK293 cells were described previously [15]. WT and *TRPM7* KO HEK293 cells were maintained in DMEM supplemented with 10% FBS, 100 µg/mL streptomycin, and 100 U/mL penicillin (all from Merck, Darmstadt, Germany). WT and *TRPM7* KO HeLa cells were acquired from Abcam (Cambridge, UK; ab265480). The frame-shift mutation in exon 14 of *TRPM7* was introduced using CRISPR/Cas9. WT and *TRPM7* KO HeLa cells were maintained in DMEM supplemented with 10% FBS, 100 µg/mL streptomycin, and 100 U/mL penicillin (all from Merck, Darmstadt, Germany). MCF-7, MDA-MB-231, and SKBR3 cells (all from DSMZ-German Collection of Microorganisms and Cell Cultures, Brunswick, Germany) were maintained in RPMI 1640 medium containing 10% FBS, 100 U/mL penicillin, 100 µg/mL streptomycin (all from Merck, Darmstadt, Germany).

### 2.4. Genome-Wide Transcriptome Profiling and qRT-PCR Analysis

To study the impact of *TRPM7* KO mutations and NS8593 on transcriptome of HAP1 cells, WT, *TRPM7* KO-01, and KO-04 cells were maintained in 25 cm^2^ flasks (Sarstedt, Nümbrecht, Germany) with IMDM containing 10% fetal bovine serum (FBS), 100 U/mL penicillin, and 100 µg/mL streptomycin (all from Thermo Fisher Scientific, Waltham, MA, USA) supplemented with 10 mM MgCl_2_ (IMDM-Mg). At ~50% confluence, the IMDM-Mg was exchanged for the regular IMDM. The cells were cultured for an additional 24 h. To extract total RNA, the medium was removed, and the cells were immediately exposed to a lysis solution followed by RNA purification using the GenElute Mammalian Total RNA Miniprep kit (Merck, Darmstadt, Germany).

To investigate the effect of NS8593, WT HAP1 cells were maintained in IMDM-Mg (to match conditions used for *TRPM7* KO cells). Next, the cells were incubated in regular IMDM containing NS8593 for 24 h, and total RNA was extracted as described above. Whole genome profiling was performed using a GeneChip Human Gene 1.0 ST Array (Affymetrix, Santa Clara, CA, USA) at Source Bioscience (Nottingham, UK) as described previously [12,39]. Processing of the array data, quality assessment, background correction, and normalization were performed with the Affymetrix Expression Console (version 1.4.0). Differential expression analysis was performed with DNASTAR ArrayStar 11.0 software (Appendix A). DNASTAR ArrayStar 11.0 software was also used to generate Venn diagrams for transcripts showing ≥ 1.5-fold changes in samples from *TRPM7* KO and NS8593-treated WT cells as compared to WT HAP1 cells and to select transcripts commonly affected in KO-01, KO-04, and NS8593-treated WT HAP1 cells versus WT cells (Appendix A). Microarray data were deposited in NCBI Gene Expression Omnibus (GEO) (GSE203013).

For qRT-PCR analysis, RNA was extracted with GenElute Mammalian Total RNA Miniprep Kit (Merck, Darmstadt, Germany). The PCR primer pairs (Metabion, Planegg, Germany) used are shown in Appendix A. The first strand cDNA synthesis was generated using the Thermo Scientific RevertAid H Minus First Strand cDNA Synthesis Kit (Thermo Fisher Scientific, Waltham, MA, USA). qPCR reactions were performed using the Thermo Scientific Absolute qPCR SYBR Green Mix (Thermo Fisher Scientific, Waltham, MA, USA) and a LightCycler 480 (Roche, Basel, Switzerland) with the following PCR settings: 95 °C 15′, 95 °C 15′′, 60 °C 15′′, 72 °C 30′′. The cycle thresholds (CT) of the test and reference (*HPRT*) genes were calculated using LightCycler 480 software (version 1.5.0, Roche, Basel, Switzerland). The relative mRNA expression levels were calculated using the 2^−ΔΔCT^ approach.

### 2.5. siRNA Targeting of TRPM7

HEK293 and SKBR3 cells were seeded in 6-well plates (Sarstedt, Nümbrecht, Germany) (~10^5^ cells/well). The next day, the cells were transiently transfected with 20 nM FlexiTube siRNAs silencing *TRPM7*: Hs_*TRPM7*_7 FlexiTube siRNA (siRNA #7), Hs_*TRPM7*_8 FlexiTube siRNA (siRNA #8), and AllStars Negative Control siRNA (all from QIAGEN, Hilden, Germany) using Lipofectamine 2000 reagent (Thermo Fisher Scientific, Waltham, MA, USA). The cells were analyzed 72 h after transfection.

### 2.6. Electrophysiological Techniques

Patch-clamp experiments were performed and analyzed as reported previously [4,24,28,59]. Briefly, whole-cell currents were recorded using an EPC10 patch-clamp amplifier (Harvard Bioscience, Holliston, MA, USA) and PatchMaster software (version V2x69, Harvard Bioscience, Holliston, MA, USA). Voltages were corrected for a liquid junction potential of 10 mV. Currents were elicited by voltage ramps from −100 mV to +100 mV over 50 ms applied every 2 s. The inward and outward current amplitudes were measured at −80 mV and +80 mV and were normalized to the cell size as pA/pF. The capacitance was measured using the automated capacitance cancellation function of EPC10. The standard extracellular solution contained 140 mM NaCl, 2.8 mM KCl, 3 mM CaCl_2_, 10 mM HEPES-NaOH, and 11 mM glucose (all from Merck, Darmstadt, Germany). Solutions were adjusted to pH 7.2 using an FE20 pH meter (Mettler Toledo, Columbus, OH, USA) and to 290 mOsm using a Vapro 5520 osmometer (Wescor Inc, South Logan, UT, USA). Patch pipettes were made of borosilicate glass (Science Products, Hofheim, Germany) and had a resistance of 2.0−3.7 MΩ when filled with the standard intracellular pipette solution containing 120 mM Cs-glutamate, 8 mM NaCl, 10 mM Cs-EGTA, 5 mM Cs-EDTA, and 10 mM HEPES-CsOH (all from Merck, Darmstadt, Germany). The intracellular solution was also adjusted to pH 7.2 and 290 mOsm.

### 2.7. Aequorin-Based Ca^2+^ Influx Assay

Measurements of [Ca^2+^]_i_ in TRPM7 expressing cells were performed and analyzed as reported previously [4,28]. HEK293 cells were maintained at 37 °C and 5% CO_2_ in DMEM supplemented with 10% FBS, 100 µg/mL streptomycin, and 100 U/mL penicillin (all from Merck, Darmstadt, Germany). Cells cultured in 6-well plates (~60% confluence) were transfected with 2 µg/dish *Trpm7* plasmid DNA and 0.1 µg/dish *pG5A* plasmid DNA encoding eGFP fused to *Aequorea victoria* aequorin, using Lipofectamine 2000 (Thermo Fisher Scientific, Waltham, MA, USA). Twenty-four hours after transfection, the cells were washed with Mg^2+^-free HEPES-buffered saline (Mg^2+^-free HBS) containing 150 mM NaCl, 5.4 mM KCl, 0.5 mM CaCl_2_, 5 mM HEPES (pH 7.4), and 10 mM glucose, and mechanically resuspended in the Mg^2+^-free HBS. For reconstitution of aequorin, cell suspensions were incubated with 5 µg/mL coelenterazine (Carl Roth, Karlsruhe, Germany) in the Mg^2+^-free HBS for 30 min at room temperature. Cells were washed twice by centrifugation at 2000 rpm for 5 min (Heraeus Pico 17 microcentrifuge, Thermo Fisher Scientific, Waltham, MA, USA), resuspended in the Mg^2+^-free HBS and aliquoted into 96-well plates (1 × 10^5^ cells per well). Luminescence was detected at room temperature using a CLARIOstar microplate reader (BMG LABTECH GmbH, Ortenberg, Germany). To monitor the effects of CP724714 and NS8593, the extracellular concentration of Ca^2+^ was increased to 5 mM by injecting the CaCl_2_-containing Mg^2+^-free HBS in the absence or presence of the inhibitors. The experiments were terminated by lysing cells with 0.05% (*v*/*v*) Triton X-100 in the Mg^2+^-free HBS to record the total bioluminescence. The bioluminescence rates (counts/s) were analyzed at 1-s intervals and calibrated as [Ca^2+^]_i_ values using the following equation: p[Ca^2+^]_i_ = 0.332588 (−log(*k*)) + 5.5593(1)
where *k* represents the rate of aequorin consumption, i.e., counts/s divided by the total number of counts.

### 2.8. Assessment of Cell Viability

To study the impact of Mg^2+^ supplementation on the growth rate of WT and *TRPM7* KO HAP1 cells (Figure 1A), the cells of each genotype were seeded in 6-well plates (10^5^ cells/well) in a standard culture medium supplemented with 10 mM MgCl_2_. After 24 h, the cell culture medium was replaced by the fresh medium with or without additional Mg^2+^, and the cell counts were determined at 24 h intervals using a Neubauer chamber (Marienfeld Superior, Lauda-Königshofen, Germany). The cell density at day 1 was accounted as 100%.

To examine the antiproliferative effects of NS8593 on HAP1 cells, the cells were seeded in a 96-well plate (5 × 10^3^ cells/well) in the standard culture medium supplemented with 10 mM MgCl_2_. After 24 h, the culture medium was replaced with the fresh medium with or without 10 mM MgCl_2_ and indicated concentrations of NS8593. The cells were further cultured for an additional 72 h. Cell Counting Kit-8 (CCK-8, Selleck Biotechnology, Frankfurt am Main, Germany) was used to determine cell density according to the manufacturer’s protocol.

To assess the antiproliferative effects of NS8593 and CP724714 on breast cancer cells, SKBR3, and MDA-MB-231 cells were seeded in 96-well plates (2 × 10^4^ cells/well) in the standard cell culture medium. After 24 h, the indicated concentrations of NS8593 and CP724714 were added to the cell culture medium and the cells were incubated for an additional 72 h. The cell density was determined using a Neubauer chamber. The synergy in the effects of NS8593 and CP724714 was assessed using the LOEWE model (Combenefit 2.021 software [60]).

### 2.9. Western Blot (WB) Analysis

To assess the effects of Zn^2+^ and zinc pyrithione on HER2 expression, HAP1 cells were seeded in a 6-well plate (10^5^ cells/well) and cultured for 72 h in the presence or absence of the supplements. After the treatment, the culture medium was removed by aspiration, and the cells were disrupted using a lysis buffer (Pierce IP Lysis Buffer, Thermo Fisher Scientific, Waltham, MA, USA; #87787) containing protease and phosphatase inhibitor cocktails (Selleck Biotechnology, Frankfurt am Main, Germany). Aliquots of the cell lysates were mixed (1:1) with 2 x *Laemmli* buffer, heated at 70 °C for 10 min and cooled on ice. The samples were separated by SDS-PAGE (4–20% gradient acrylamide/bis-acrylamide, Bio-Rad, Feldkirchen, Germany) and electroblotted on nitrocellulose membranes (Merck, Darmstadt, Germany; #GE10600002). After blocking with 5% (*w*/*v*) non-fat dry milk in Tris-buffered saline with 0.1% Tween 20 (TBST), the upper part of the membrane was probed by a rabbit monoclonal anti-HER2 antibody (Cell Signaling Technology, Danvers, MA, USA; #2165, 1:2000) in TBST with 5% (*w*/*v*) bovine serum albumin (BSA; Merck, Darmstadt, Germany) overnight at 4 °C, followed by triple washing steps in TBST, incubation with a horseradish peroxidase-coupled anti-rabbit lgG (Cell Signaling Technology, Danvers, MA, USA; #7074V, 1:1000) in TBST with 5% (*w/v*) BSA for 1 h at r.t. and washing in TBST. The lower part of the membrane was assessed by the anti-mouse monoclonal anti-β-Actin-Peroxidase (Merck, Darmstadt, Germany; #A3854, 1:100,000) in TBST with 5% (*w*/*v*) BSA for 1 h at r.t. and triple washing steps in TBST. Blots were examined by the luminescence imager ChemiDoc (Bio-Rad, Feldkirchen, Germany). The HER2 signal was normalized to the signal obtained with the mouse monoclonal anti-β-Actin-Peroxidase (Merck, Darmstadt, Germany; #A3854, 1:100,000) using the Image Studio Lite Ver 5.2 software. WB analysis of HER2 expression in other cells was conducted similarly.

### 2.10. Statistical Analysis

Data are presented as the means ± standard deviation (SD). Data showed a normal distribution. Data were compared using a two-tailed *t*-test. For multiple comparisons, an ANOVA (GraphPad Prism 10.1.2 software) was used. Significance was accepted at *p ≤* 0.05.

## 3. Results

### 3.1. Genetic and Pharmacological Inactivation of TRPM7 Suppresses the Proliferation of Human Leukemia HAP1 Cells

In previous studies, we employed a human leukemia HAP1 cell line (referred to as clone KO-04) carrying a frame-shift mutation in the *TRPM7* gene [12,39]. Because clonal selection can produce a considerable diversity of cell phenotypes, we took advantage of an alternative HAP1 cell line with a frame-shift mutation in *TRPM7* (clone KO-01) (Appendix A). TRPM7 functions as a constitutively active cation channel, which is negatively regulated by intracellular Mg^2+^ through a regulatory site located in the lower channel gate of TRPM7 [2,59]. In patch-clamp measurements, the induction of whole-cell TRPM7 currents is achieved by removing cytosolic Mg^2+^ using EDTA-containing patch pipette solutions [2]. We applied this approach to characterize HAP1 cells. Patch-clamp measurements with parental WT HAP1 cells (clone WT-C631) revealed characteristic endogenous TRPM7 currents, which were entirely abrogated in both *TRPM7* knockout (KO) cell lines (Figure 1A).

Because frame-shift mutations in *TRPM7* permanently eliminate channel and kinase activities of TRPM7, we asked whether NS8593, a potent inhibitor of the TRPM7 channel [29,35], can serve as an alternative strategy to block the TRPM7 channel in HAP1 cells acutely. We observed that exposure of WT HAP1 cells to 10 or 30 µM NS8593 markedly suppressed the TRPM7 currents (Figure 1B).

In accordance with previous studies [10,11,12], we found that *TRPM7* KO HAP1 cells grow normally in a cell culture medium containing an additional 10 mM Mg^2+^. However, we observed that maintaining *TRPM7* KO HAP1 cells in a regular cell culture medium suppressed the proliferation of KO-01 and KO-04 cell lines and that this effect was well noticeable after 48 h of the medium’s replacement (Figure 1C,D). Next, we cultured WT HAP1 cells in the regular cell culture medium containing different concentrations of NS8593 for 72 h and observed concentration-dependent inhibition of cell growth (Figure 1E). However, the proliferation of NS8593-treated HAP1 cells was fully normalized in the Mg^2+^-supplemented medium (Figure 1E).

Hence, the pharmacological inhibition of the TRPM7 channel replicated the effects of *TRPM7* KO mutations on endogenous TRPM7 currents and the proliferation of HAP1 cells.

### 3.2. Gene Expression Profiling of HAP1 Cells Reveals New Regulatory Targets of TRPM7

Previously, we conducted genome-wide transcriptome profiling of mouse tissues to investigate organismal responses upon deleting the *Trpm6* and *Trpm7* genes in mice [12,39]. In the present study, we applied this approach to uncover cellular pathways affected by the deletion of the *TRPM7* gene in HAP1 cells. We cultured the parental WT HAP1 cells and two *TRPM7* KO clones in the regular cell culture medium (without additional 10 mM Mg^2+^) for 24 h, the incubation time preceding the growth defect of *TRPM7* KO cells (Figure 1C,D). Consequently, RNA was acutely extracted from adherent cells, and transcriptome profiling was performed analogously to our previous studies [12,39]. Applying a cut-off value of 1.5-fold changes, we identified 274 up- and 201 down-regulated genes in *TRPM7* KO-01 cells compared to WT HAP1 cells (Figure 2A, Appendix A). A similar analysis of *TRPM7* KO-04 cells revealed 473 up- and 221 down-regulated genes (Figure 2A, Appendix A). Among other transcripts, we noted that the receptor tyrosine-protein kinase *HER2* gene (*ERBB2*) expression was remarkably up-regulated in both *TRPM7* KO clones (Appendix A). Interestingly, transcripts of α-kinases eEF-2K and ALPK1–3 were detected in HAP1 cells (Appendix A), but the expression levels of only *ALPK1* were increased in either of the *TRPM7* KO cell lines (Appendix A).

Next, we investigated how inhibition of the TRPM7 channel by NS8593 could impact the transcriptome of WT HAP1 cells and whether these changes can be correlated to the effects of *TRPM7* KO mutations. To this end, we cultured WT HAP1 cells for 24 h in the regular cell culture medium containing 30 µM NS8593, followed by transcriptome profiling analogously to the experiment with *TRPM7* KO cells. This analysis identified 228 up- and 111 down-regulated genes in NS8593-treated cells compared to WT HAP1 cells (Figure 2A, Appendix A). Next, we compared these results with two datasets obtained with *TRPM7* KO cells. We found that pharmacological or genetic ablation of TRPM7 activity in HAP1 cells caused up-regulation of 32 genes and down-regulation of 12 genes (Figure 2A, Appendix A). Hence, the combinatory profiling of HAP1 cells enabled us to narrow the list of putative regulatory targets of *TRPM7* for future analysis.

We noted that NS8593-treated WT HAP1 cells increased the expression of *HER2* and *ALPK1*, hence recapitulating the impact of *TRPM7* KO mutations (Appendix A). These findings were re-examined by the qRT-PCR approach, which demonstrated that the expression levels of *HER2*, but not *HER3*, were significantly higher in the two *TRPM7* KO cell lines (Figure 2B,C). The exposure of WT HAP1 cells to NS8593 also increased *HER2* expression, but these changes were not statistically significant in these settings (Figure 2B). *ALPK1* expression was similarly up-regulated in NS8593-treated WT and *TRPM7* KO cells (Figure 2D). 

### 3.3. TRPM7 Regulates Expression Levels of HER2 in a Zn^2+^-Dependent Fashion

The crucial role of HER2 in the diagnosis and treatment of breast cancer [61] prompted us to study how TRPM7 regulates the expression of *HER2*. Using the western blotting approach, we found that HER2 protein levels were 5.0- and 2.5-fold higher in TRPM7 KO-01 and KO-04 cells than in parental WT HAP1 cells (Figure 3A). Since the *TRPM7* KO-04 HAP1 cells were extensively studied previously [12,35,39,62], we selected this cell line for further experiments.

The TRPM7 channel is highly permeable for divalent cations, including Zn^2+^, and its genetic deletion leads to the cellular deprivation of Zn^2+^ in *TRPM7* KO-04 HAP1 cells [12]. However, Zn^2+^ levels were normalized in *TRPM7* KO-04 HAP1 cells after Zn^2+^ supplementation, arguing that alternative Zn^2+^ transporters can compensate for the lack of TRPM7 [12]. In accordance with this assumption, we showed previously that the uptake of radioactive ^65^Zn^2+^ was significantly reduced but not completely abolished in *TRPM7* KO-04 HAP1 cells [12]. Along these lines, we investigated herein whether adding Zn^2+^ to the cell culture medium can reverse the effect of *TRPM7* KO-04 cells on the expression of HER2. We observed that adding 10 µM Zn^2+^ to the regular medium could ameliorate the impact of *TRPM7* KO on HER2 expression (Figure 3B). The expression of HER2 was further normalized in *TRPM7* KO-04 cells after the application of 0.5 µM zinc pyrithione (ZP), a broadly used Zn^2+^ ionophore (Figure 3C). In agreement with these findings, analogous treatment of WT HAP1 cells by additional Zn^2+^ or ZP caused a significant reduction of HER2 expression (Figure 3D). These results support the notion that deletion of *TRPM7* indirectly altered HER2 expression, likely due to Zn^2+^ deprivation of HAP1 cells.

We noted that HER2 expression was not altered in WT HAP1 cells cultured for 24 h in the presence of NS8593 (Figure 3A). We asked whether a more prolonged inhibition of the TRPM7 channel is necessary to replicate the impact of *TRPM7* KO mutations on HER2 expression. To this end, we exposed WT HAP1 cells to NS8593 for 48–72 h and found that the prolonged application of the TRPM7 inhibitor caused a gradual increase in HER2 levels (Figure 3E).

The haploid HAP1 cells were engineered from cells isolated from a patient with chronic myeloid leukemia [63]. To investigate whether HER2 is the regulatory target of TRPM7 in cells of non-hematopoietic origin, we examined HER2 expression in embryonic trophoblast stem (TS) cells isolated from WT and *Trpm7* KO mice [28,39]. Unexpectedly, we found that *Trpm7* KO TS cells displayed significantly reduced HER2 expression levels compared to WT TS cells. This effect was partially reversed by adding Zn^2+^ or ZP to the cell culture medium (Figure 4A). Furthermore, the exposure of WT TS cells to the medium supplemented by Zn^2+^ or ZP increased the expression of HER2 (Figure 4B).

HEK293 cells were often used to study the cellular role of endogenous TRPM7 [15,24,28,64]. We found that, analogously to TS cells, HEK293 cells with *TRPM7* KO mutation [15] displayed reduced levels of HER2 compared to WT cells and that Zn^2+^ supplementation ameliorated this phenotype (Figure 4C). Consistently, the exposure of WT HEK293 cells to the Zn^2+^-enriched medium resulted in up-regulation of HER2 expression (Figure 4E). However, we noted that the application of ZP did not normalize HER2 levels in WT and *TRPM7* KO HEK293 cells (Figure 4D,E). Next, we asked whether siRNA silencing of TRPM7 could alter HER2 expression analogously to the *TRPM7* KO mutation. Applying two alternative TRPM7-specific siRNA constructs, we observed that both treatments lowered HER2 levels in HEK293 cells and that this effect was comparable to the impact of *TRPM7* KO mutation (Figure 4F).

In addition, we studied the impact of *TRPM7* KO on human HeLa cells, which were derived from cervical cancer [65]. Using the patch-clamp approach, we found that WT HeLa cells display characteristic TRPM7 currents, which were fully diminished in *TRPM7* KO cells (Appendix A). Analogously to our experiments with TS and HEK293 cells, *TRPM7* KO caused down-regulation of HER2 expression in HeLa cells (Appendix A). However, we observed that Zn^2+^ supplementation of *TRPM7* KO HeLa cells did not normalize HER2 levels, likely reflecting the inability of alternative Zn^2+^ transporter proteins to compensate for the role of TRPM7 in these cells (Appendix A). In line with this notion, adding Zn^2+^ to the culture medium of WT HeLa cells caused only modest up-regulation of HER2 expression, but these changes were not statistically significant (Appendix A).

Human breast cancer SKBR3 cells expressing very high levels of HER2 are frequently used for testing new therapeutic strategies. Using western blotting, we could confirm that SKBR3 cells express remarkably high HER2 levels compared to other breast cancer MCF-7 and MDA-MB-231 cells (Figure 5A). Next, we investigated whether HER2 is the regulatory target of TRPM7 in SKBR3 cells. Analogously to other cells (Figure 4, Appendix A), we observed that two alternative anti-*TRPM7* siRNAs caused significant down-regulation of HER2 levels in SKBR3 cells (Figure 5B). In addition, we performed patch-clamp measurements of endogenous TRPM7 currents in SKBR3 and MDA-MB-231 (Figure 5C). We found that both cell lines displayed similar current amplitudes, suggesting that increased expression of HER2 did not affect the TRPM7 channel activity (Figure 5C).

Taken together, our experiments with various cell lines revealed that TRPM7 regulates the expression levels of HER2 protein in a Zn^2+^-dependent fashion.

### 3.4. Combinatory Pharmacological Inhibition of TRPM7 and HER2 Elicits the Synergistic Effect on the Proliferation of HER2-Positive Breast Cancer Cells

It is well-documented that genetic or pharmacological inactivation of TRPM7 causes an inhibitory effect on the growth of tumor cells (reviewed in [25]). In accordance with this notion, we observed that NS8593 suppressed the proliferation of SKBR3 and MDA-MB-231 cells in a concentration-dependent fashion (Figure 5D). Moreover, the treatment of SKBR3 cells by NS8593 significantly reduced HER2 levels (Figure 5E).

Consequently, we asked whether a combined pharmacological inhibition of TRPM7 and HER2 could improve the treatment of HER2-positive cancer cells. We exposed SKBR3 cells to the potent HER2 inhibitor CP724714 [66] and found that this compound dose-dependently suppressed the growth of SKBR3 cells (Figure 6A). Co-administration of 10 or 20 µM NS8593 with CP724714 further suppressed the proliferation of SKBR3 cells (Figure 6B,C). Applying a LOEWE analysis [60,67], we established that a combinatory treatment with 10 µM NS8593 and 0.5 µM CP724714 synergistically affected cell growth (Figure 6D). Furthermore, adding 20 µM NS8593 elicited the synergetic effect in the presence of the whole range of CP724714 concentrations (Figure 6D). To rule out that the synergetic effect of CP724714 was elicited due to off-target inhibition of TRPM7, we examined transiently expressed TRPM7 in WT HEK293 cells and examined TRPM7 activity in the presence of either 10 µM CP724714 or 10 µM NS8593. We observed that NS8593 inhibited the channel, whereas CP724714 showed no effects on TRPM7 (Appendix A).

Finally, we conducted analogous experiments with MDA-MB-231 cells expressing HER2 at a low level (Appendix A). MDA-MB-231 cells were insensitive to the relevant concentrations of CP724714. Moreover, CP724714 could not enhance the antiproliferative effects of 5 and 20 µM NS8593 and rather weakened the impact of 10 µM NS8593 (Appendix A). These results support the idea that the synergistic action of NS8593 and CP724714 on SKBR3 cells was attributed to the high expression of HER2.

## 4. Discussion

In the present study, we conducted transcriptome profiling of human leukemia HAP1 cells to get insights on cellular responses triggered by inhibition of the TRPM7 channel using the pharmacological agent NS8593. Among other entities, expression levels of HER2 (ERBB2) were found to be affected by either NS8593 or genetic disruption of *TRPM7* in HAP1 cells and other cell lines. Given the exceptional importance of HER2 for treating breast cancer, we performed proof-of-principle experiments to show that combined pharmacological inhibition of HER2 and TRPM7 elicits synergistic antiproliferative effects on HER2-positive breast cancer cells.

HAP1 cells represent a near-haploid cell line engineered from cells isolated from a patient with chronic myeloid leukemia triggered by the *BCR-ABL* fusion mutation [63]. As HAP1 cells only have one copy of each gene, the CRISPR/Cas9 technology allows a rapid introduction of loss-of-function mutations in these cells [63]. Consequently, HAP1 cells were widely used in various biomedical studies, including functional profiling of oncogenic mutations and genome-wide screens for new anti-cancer targets [68,69,70,71]. More recently, our group and other laboratories explored HAP1 cells to elucidate the cellular role of TRPM7 [12,35,39,62,72]. Thus, we examined the HAP1 cell line carrying a loss-of-function mutation in the *TRPM7* gene to demonstrate that the TRPM7 channel regulates cellular levels of Zn^2+^ [12,39]. In the present study, we investigated the impacts of the *TRPM7* KO mutation and TRPM7 inhibitor NS8593 on the transcriptome of HAP1 cells. Such comparative analysis revealed that TRPM7 regulates HER2 expression in a Zn^2+^-dependent manner [12,15].

The clinical relevance of HER2 for targeting breast cancer therapy is well-documented [61]. Up to 30% of breast cancers overexpress HER2, whereby high HER2 expression is associated with a more aggressive disease progression and higher recurrence rate [61]. Many patients with HER2-overexpressing cancer respond to HER2 inhibitors such as trastuzumab, pertuzumab, lapatinib, and neratinib [73]. However, patients frequently develop drug resistance through different mechanisms, for instance, acquiring mutations in HER2 or compensatory changes in cell signaling [73,74]. To this end, the identification of new pathways connected to HER2 can offer alternative strategies for combinatorial drug treatment of breast cancer [73].

The present study investigated the TRPM7/HER2 relationship in various cells, including embryonic TS cells, HEK293, HeLa, MDA-MB-231, and SKBR3 cells. These experiments demonstrated that TRPM7 regulates the expression of HER2. However, unlike in HAP1 cells, the deactivation of TRPM7 in TS cells, HEK293, HeLa, MDA-MB-231, and SKBR3 cells resulted in the down-regulation of HER2. We noted that another group investigated the impact of *Trpm7* KO on the transcriptome of mouse embryonic stem (ES) cells and reported that *HER2* was down-regulated in *Trpm7* KO ES cells [16]. Previously, we performed genome-wide transcriptome profiling of villi isolated from the whole intestine of mice with enterocyte-specific *Trpm7* KO [12]. Intriguingly, *HER2* was down-regulated in *Trpm7*-deficient villi [12]. The distinct response of HAP1 cells may be attributed to the hematopoietic origin of these cells associated with distinct Zn^2+^ signaling and expression profiles of Zn^2+^ finger proteins [75]. In addition, a compensatory impact of Zn^2+^ transporters on the *TRPM7* KO phenotype could play a role. Apart from TRPM7, twenty-four members of the solute carriers of family 30 (Slc30a1–10 or ZnT1–10) and family 39 (Slc39a1–14 or Zip1–14) were proposed to orchestrate cellular Zn^2+^ balance in a cell-specific mode [76]. It is worth noting that analysis of human breast cancer tissues showed significant Zn^2+^ accumulation, especially in HER2-positive and triple-negative cancers and that this effect correlated with the grade of malignancy [77,78]. In another study, experiments with MCF-7 breast cancer cells demonstrated that TRPM7 regulates cellular levels of MDMX by modulating the intracellular Zn^2+^ levels [79]. MDMX is a zinc-containing negative regulator of p53, which is overexpressed in various cancers and implicated in cancer initiation and progression [79]. Along these lines, further studies are necessary to establish the mechanisms of the Zn^2+^-dependent transcriptional regulation of HER2.

NS8593 was initially characterized as an inhibitor of Ca^2+^-activated K^+^ (K_Ca_2.1–2.3) channels [80]. Further research revealed that NS8593 is also a potent inhibitor of the TRPM7 channel [81]. In non-excitable cells, K_Ca_2.1–2.3 channels hyperpolarize the plasma membrane, thereby enhancing the influx of divalent cations [80]. On the other hand, the TRPM7 channel controls the cellular uptake of Zn^2+^, Ca^2+^, and Mg^2+^ [2,12]. Therefore, the dual specificity of NS8593 could be advantageous for treating rapidly growing cancer cells, which heavily rely on a steady supply of divalent cations and other nutrients [10,20,35,37,38,39,40,41]. In the present work, we investigated whether NS8593 can be advantageous for the combined targeting of HER2-positive cancer cells. Our proof-of-concept experiments showed that the co-application of pharmacological inhibitors of HER2 and TRPM7 had a synergistic antiproliferative effect on HER2-positive breast cancer SKBR3 cells but not HER2-deficient MDA-MB-231 cells. Collectively, our results reinforced the idea that TRPM7 represents a promising therapeutic target for treating breast cancer [38,46,47,48,49,50,51].

## 5. Conclusions

Transcriptome profiling of human leukemia HAP1 cells revealed that TRPM7 plays a regulatory role in the expression of several transcripts, including *HER2* (*ERBB2*). Further examination of HAP1 cells and several non-hematopoietic cells demonstrated that TRPM7 affects the expression of *HER2* in a Zn^2+^-dependent fashion. Co-inhibition of HER2 and TRPM7 elicits a synergistic antiproliferative effect on HER2-overexpressing SKBR3 cells. Collectively, our findings suggest a new combinatorial approach for a targeted therapy for HER2-positive breast cancer.

## Figures and Tables

**Figure 1 cells-13-01801-f001:**
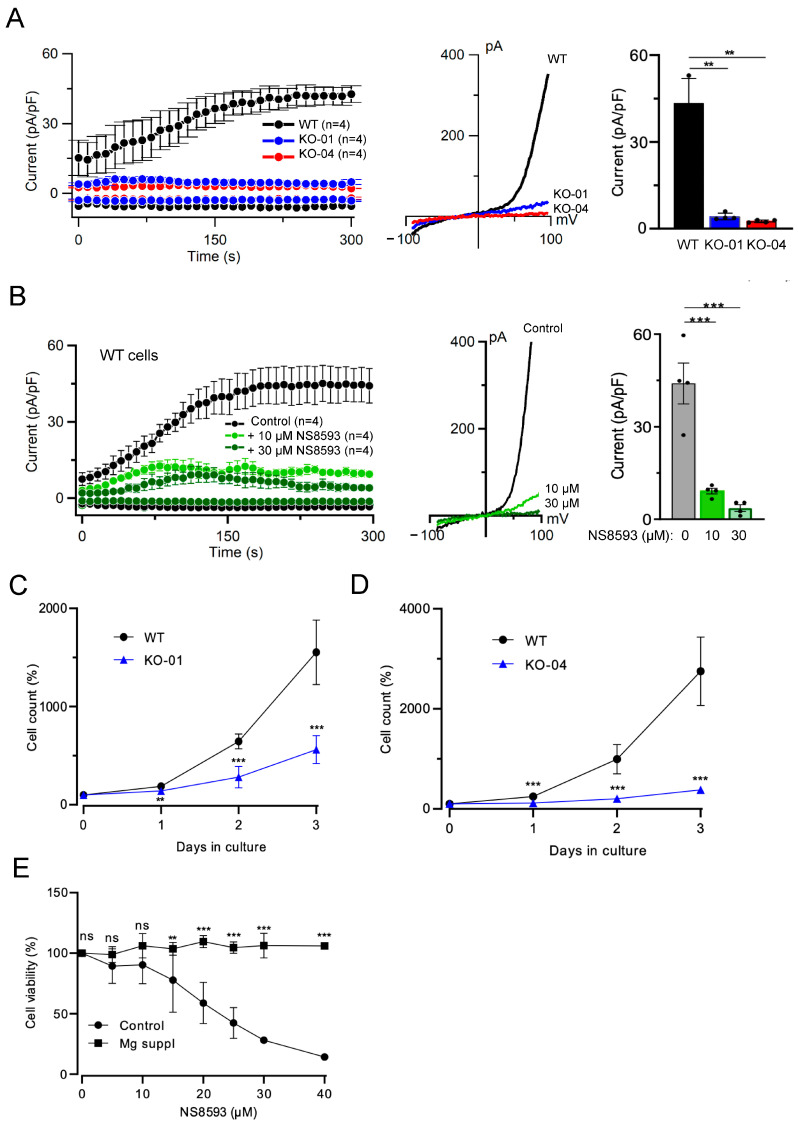
Assessment of human leukemia HAP1 cells. (**A**) Left panel: Whole-cell currents measured at −80 and +80 mV over time in WT (WT) and two *TRPM7* KO (KO-01 and KO-04) HAP1 cell lines. Middle panel: Representative current−voltage (I−V) relationships obtained at 300 s in measurements illustrated on the Left panel. Right panel: Bar graphs of current amplitudes at +80 mV (300 s) illustrated on the Left panel. Data are mean ± SD; *n*, the number of cells examined. ** *p* ≤ 0.01 (one-way ANOVA). (**B**) Left panel: Whole-cell currents measured at −80 and +80 mV over time in WT HAP1 cells in the absence (Control) and the presence of 10 or 30 µM NS8593. Middle panel: Representative I−V relationships obtained at 300 s in measurements illustrated on the Left panel. Right panel: Bar graphs of current amplitudes at +80 mV (300 s) illustrated on the Left panel. Data are mean ± SD; *n*, the number of cells examined. *** *p* ≤ 0.001 (one-way ANOVA). (**C**,**D**) Proliferation rate of WT and *TRPM7* KO-01 (**C**) and KO-04 (**D**) HAP1 cells. The cells were cultured for 3 days in the regular cell culture medium. The initial cell density (Day 0) was accounted as 100%. Data are mean ± SD of *n* = 3 independent experiments. *** *p* ≤ 0.001; ** *p* ≤ 0.01 (*t*-test). (**E**) Viability of WT HAP1 cells maintained in regular cell culture medium (Control) or medium with an additional 10 mM MgCl_2_ (Mg suppl) containing different concentrations of NS8593 for 72 h. Cell densities in the absence of NS8593 were accounted as 100%. Data are mean ± SD of *n* = 3 independent experiments. *** *p* ≤ 0.001; ** *p* ≤ 0.01; ns—not significantly different (*t*-test).

**Figure 2 cells-13-01801-f002:**
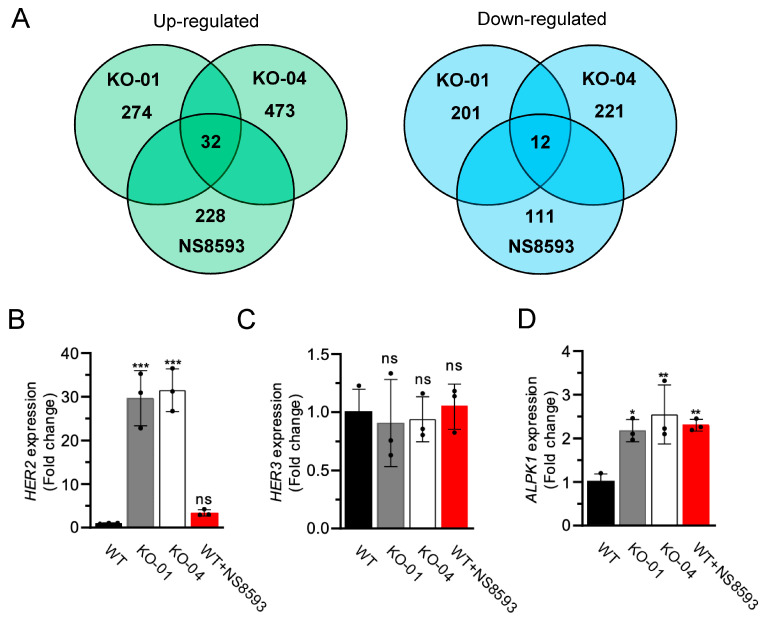
Genome-wide transcriptome profiling of HAP1 cells. (**A**) Venn diagrams for transcripts showing ≥1.5-fold up-regulation (Left panel) and down-regulation (Right panel) in *TRPM7* KO-01 (KO-01), *TRPM7* KO-04 (KO-04), and NS8593-treated WT (NS8593) HAP1 cells as compared to untreated WT HAP1 cells. (**B**–**D**) Relative expression levels of *HER2 (ERBB2)* (**B**) and *HER3 (ERBB3)* (**C**) and *ALPK1* (**D**), assessed by qRT-PCR approach in WT (WT), *TRPM7* KO-01 (KO-01), *TRPM7* KO-04 (KO-04), and NS8593-treated WT (WT+NS8593) HAP1 cells with *HPRT* as a reference transcript. Data are mean ± SD of *n* = 3 independent experiments. *** *p* ≤ 0.001; ** *p* ≤ 0.01; * *p* ≤ 0.05; ns—not significantly different (one-way ANOVA).

**Figure 3 cells-13-01801-f003:**
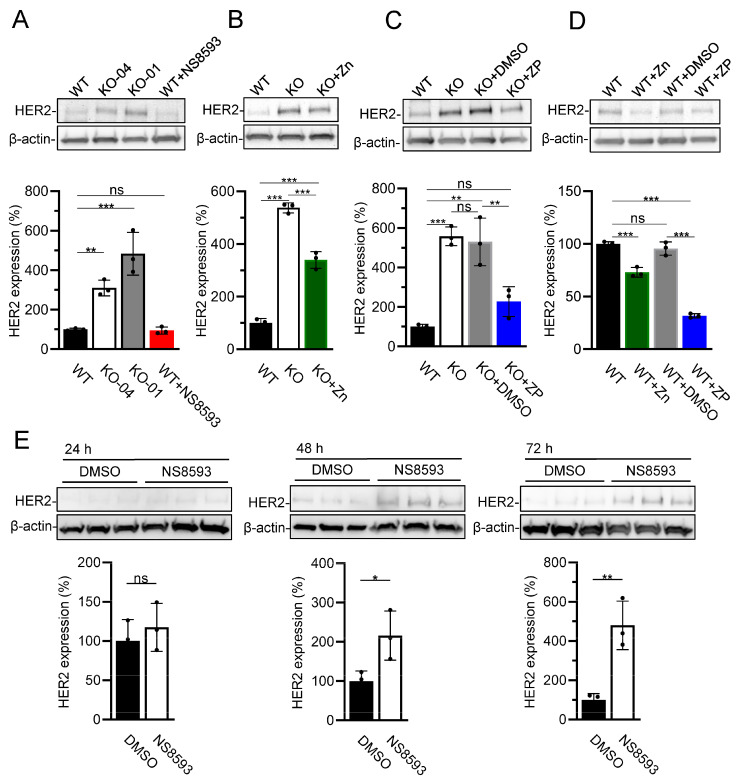
The impact of TRPM7 on HER2 expression levels in HAP1 cells. (**A**) HER2 expression in WT (WT), *TRPM7* KO-04 (KO-04), *TRPM7* KO-01 (KO-01), and NS8593-treated WT (WT+NS8593) HAP1 cells. (**B**–**D**) Effects of 10 µM ZnCl_2_ (Zn) and 0.5 µM zinc pyrithione (ZP) on HER2 expression in WT (WT) and *TRPM7* KO-04 (KO) HAP1 cells. (**E**) HER2 expression in WT HAP cells treated by 20 µM NS8593 for 24, 48, and 72 h. In (**C**–**E**), equal DMSO volumes (DMSO) were used instead of ZP or NS8593. Upper panels: Representative western blots are shown. Equal volumes of cell lysates were assessed using anti-HER2 and anti-β-actin antibodies. Lower panels: Bar graphs showing normalized HER2 expression levels in experiments from the Upper panel. The ratio of HER2 and anti-β-actin signals in untreated WT HAP1 cells was accounted as 100%. The results shown in the bar graphs are mean ± SD of *n* = 3 independent experiments. In (**A**–**D**), *** *p* ≤ 0.001; ** *p* ≤ 0.01; ns—not significantly different (one-way ANOVA). In (**E**), ** *p* ≤ 0.01; * *p* ≤ 0.05; ns—not significantly different (*t*-test).

**Figure 4 cells-13-01801-f004:**
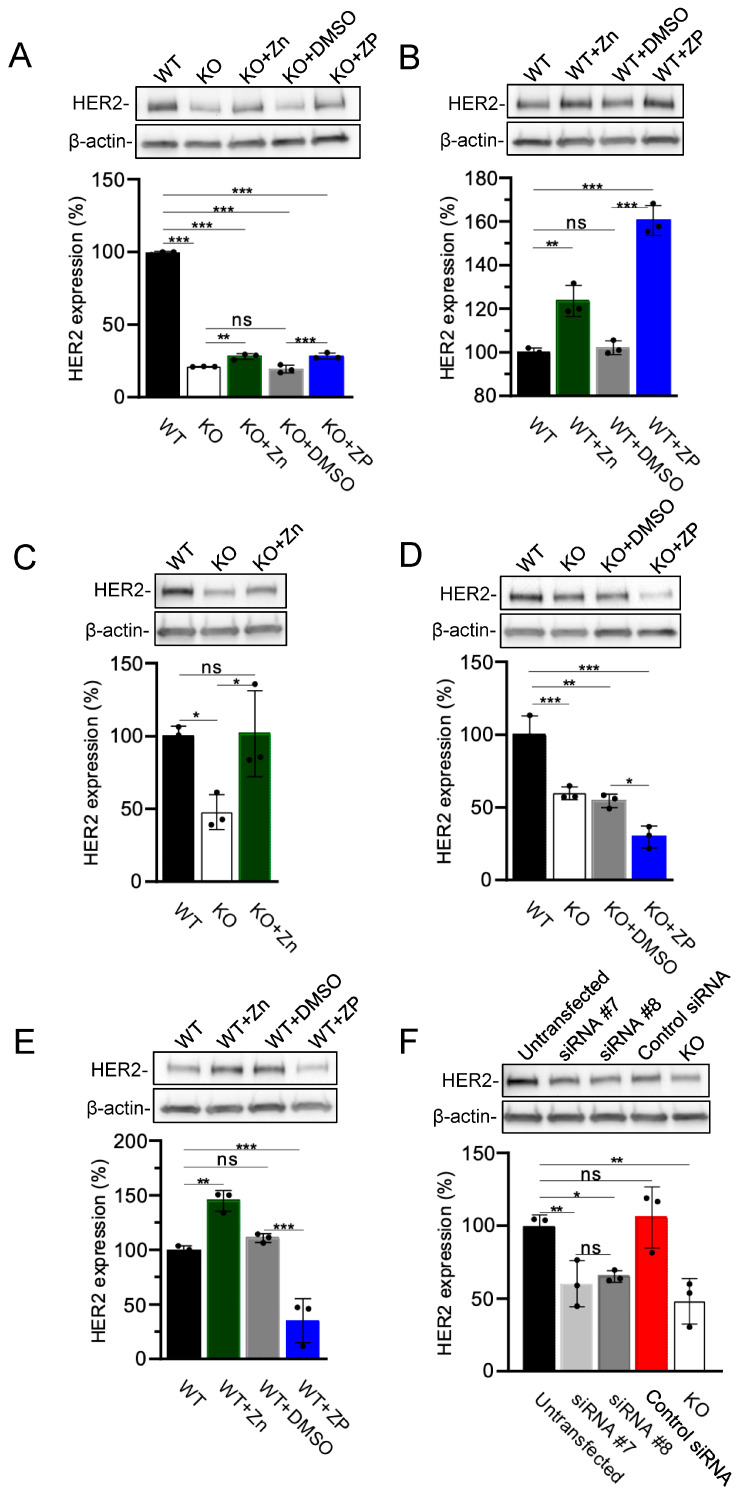
The regulatory effect of TRPM7 on HER2 expression in embryonic trophoblast stem (TS) cells and HEK293 cells. (**A**,**B**) Assessment of HER2 expression in WT (WT) and *TRPM7* KO (KO) TS cells. Upper panels: Representative western blots obtained with the cells cultured in the absence or presence of 10 µM ZnCl_2_ (Zn) and 0.5 µM zinc pyrithione (ZP) are shown. Equal volumes of cell lysates were assessed using anti-HER2 and anti-β-actin antibodies. Equal volumes of DMSO (DMSO) were used instead of ZP as an additional control. Lower panels: Bar graphs showing normalized HER2 expression levels in experiments in the Upper panels. The ratio of HER2 to anti-β-actin signals in WT TS cells was accounted as 100%. (**C**–**E**) Assessment of HER2 expression in WT (WT) and *TRPM7* KO (KO) HEK293 cells in the absence or presence of 10 µM ZnCl_2_ (Zn) and 0.5 µM zinc pyrithione (ZP). The experiments were performed and analyzed analogously to (**A**,**B**). (**F**) Analysis of HER2 expression in untransfected WT (Untransfected) HEK293 cells, WT HEK293 cells transfected by *TRPM7*-specific siRNA #7 (siRNA #7), *TRPM7*-specific siRNA #8 (siRNA #*8*) and AllStars Negative Control siRNA (Control siRNA), and *TRPM7* KO (KO) HEK293 cells. Experiments were performed and analyzed analogously to (**A**,**B**) except that the ratio of HER2 to anti-β-actin signal in untransfected WT HEK293 cells was accounted as 100%. The results in the bar graphs are mean ± SD of *n* = 3 independent experiments. *** *p* ≤ 0.001; ** *p* ≤ 0.01; * *p* ≤ 0.05; ns—not significantly different (one-way ANOVA).

**Figure 5 cells-13-01801-f005:**
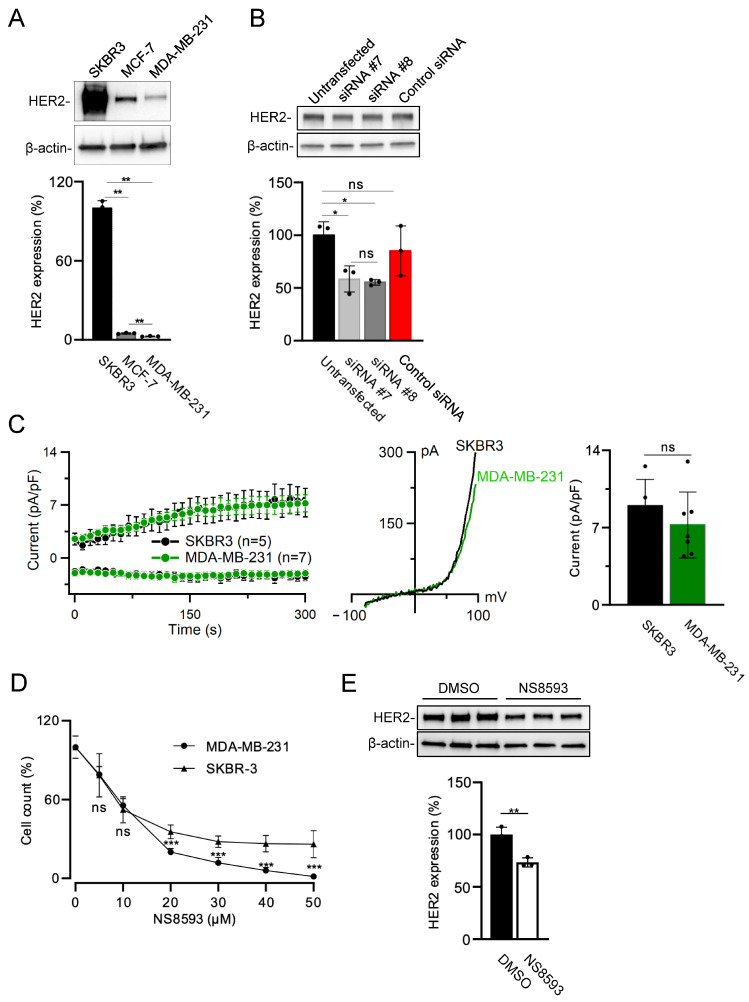
Assessment of breast cancer SKBR3, MCF-7, and MDA-MB-231 cells. (**A**) Analysis of HER2 expression in SKBR3, MCF-7, and MDA-MB-231 cells. Upper panel: Representative western blot obtained with SKBR3, MCF-7, and MDA-MB-231 cells. Equal volumes of cell lysates were assessed using anti-HER2 and anti-β-actin antibodies. Lower panel: Bar graph showing normalized HER2 expression levels in experiments from the Upper panel. The ratio of HER2 to anti-β-actin signal in SKBR3 cells was accounted as 100%. (**B**) Assessment of HER2 expression in untransfected (Untransfected) SKBR3 cells and SKBR3 cells transfected by *TRPM7*-specific siRNA #7 (siRNA #7), siRNA #8 (siRNA #8), and AllStars Negative Control siRNA (Control siRNA). Experiments were performed and analyzed analogously to (**A**) except that the ratio of HER2 to anti-β-actin signal in untransfected SKBR3 cells was accounted as 100%. The results in the bar graphs in (**A**,**B**) are mean ± SD of *n* = 3 independent experiments. ** *p* ≤ 0.01; * *p* ≤ 0.05; ns—not significantly different (one-way ANOVA). (**C**) Comparison of endogenous TRPM7 currents measured in SKBR3 and MDA-MB-231 cells. Left panel: Whole-cell currents measured at −80 and +80 mV over time in SKBR3 and MDA-MB-231 cells. Middle panel: Representative current−voltage (I−V) relationships obtained at 300 s in measurements shown in the Left panel. Right panel: Bar graph of current amplitudes at +80 mV (300 s) illustrated on the Left panel. Data are mean ± SD; *n*, the number of cells examined. Ns—not significantly different (*t*-test). (**D**) Viability of SKBR3 and MDA-MB-231 cells exposed to different concentrations of NS8593 for 72 h. Cell densities in the absence of NS8593 were accounted as 100%. Data are mean ± SD of *n* = 3 independent experiments. *** *p* ≤ 0.001; ns—not significantly different (*t*-test). (**E**) HER2 expression in SKBR3 cells treated by 30 µM NS8593 (NS8593) or equal volumes of DMSO (DMSO) for 72 h. Upper panel: Representative western blots are shown. Equal volumes of cell lysates were assessed using anti-HER2 and anti-β-actin antibodies. Lower panel: Bar graphs showing normalized HER2 expression level in experiments from the Upper panel. The results in the bar graphs are mean ± SD of *n* = 3 independent experiments. ** *p* ≤ 0.01 (*t*-test).

**Figure 6 cells-13-01801-f006:**
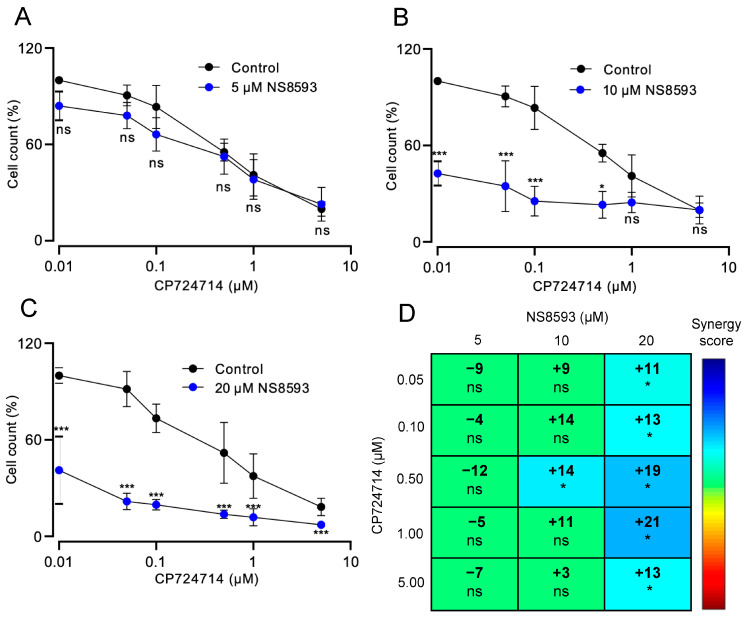
Combinatory treatment of SKBR3 cells by NS8593 and CP724714. (**A**–**C**) Viability of SKBR3 cells treated by different concentrations of CP724714 for 72 h in the absence (Control) and presence of 5 µM (**A**), 10 µM (**B**), or 20 µM (**C**) NS8593. Data are mean ± SD of *n* = 3 independent experiments. *** *p* ≤ 0.001; * *p* ≤ 0.05; ns—not significantly different (one-way ANOVA). (**D**) LOEWE synergy analysis of the cytotoxic effects elicited by NS8593 and CP724714 on SKBR3 cells in (**A**–**C**). * *p* ≤ 0.05; ns—not significantly different (one-way ANOVA).

## Data Availability

NCBI GEO–GSE203013, Whole-genome profiling of human HAP1 cells after the genetic and pharmacological inactivation of TRPM7.

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
