# Peer review of "Expression Profiling Identified TRPM7 and HER2 as Potential Targets for the Combined Treatment of Cancer Cells"

_cells, 2024, doi:10.3390/cells13211801_

Round 1
Reviewer 1 Report
Comments and Suggestions for Authors
This is a very compelling and important study investigating the impact of TRPM7 on gene expression, an area that is under explored. The authors found that HER2 (ERRB2) is upregulated by knockout of TRPM7 or inhibition of the TRPM7 channel by NS8593 in HAP1 cells. This finding was rigorously verified by qRT-PCR and Western Blotting. To investigate whether the TRPM7 KO effect on HER2 is generalizable, the authors analyzed expression of HER2 in other cellular models but found that HER2 was downregulated. Interestingly, a relationship between TRPM7, HER2, and zinc was uncovered in the authors investigation. To explore the potential clinical significance of the gene expression relationship, the authors investigated whether NS8593 and CP724714 synergistically suppress of HER2 positive breast cancer cells. Pharmacological synergism was detected, supporting the notion that pharmacological agents targeting both TRPM7 and HER2 could potentially be beneficial to the treatment of HER2 positive cancers. Overall, this is a rigorous and well-executed study. Some minor concerns regarding the manuscript are listed below.
1. Throughout the manuscript the word "regulate" is used to describe TRPM7's impact on HER2 expression. However, this relationship may be indirect and not evidence of direct regulation of HER2 gene expression by TRPM7. I would recommend softening the language (e.g., influences or affects).
2. The only supplementary files that could be downloaded were supplementary Table 1 and Table 2. Supplementary Table 3 was missing. Supplementary Figures 1-3 were not available for review.
3. Does CP724714 affect TRPM7 channel conductance? The paper would be strengthened by showing that CP724717 acts independent of TRPM7. This is important as a quite a few molecules appear to influence the channel. Please include if available.
4. In the methods section, "cytotoxicity" is used, but the authors are not looking at cell death. Please change to reflect what was being assayed (e.g., cell proliferation or cell number, etc.).
Reviewer 2 Report
Comments and Suggestions for Authors
Reviewer 3 Report
Comments and Suggestions for Authors
This study performed genome-wide transcriptome profiling of human leukemia cells to identify TRPM7 target genes using CRISPR/Cas9 knockouts of TRPM7 as well as its pharmacological inhibition by NS8593. The authors claim that TRPM7 regulates HER2 in a Zn2+-dependent manner and that co-inhibition of TRPM7 and HER2 display a synergistic anti-proliferative effect in HER2-overexpressing breast cancer cells but not in HER2-deficient breast cancer cells. The rationale of the study is solid given various reports on TRPM7’s role in cancer, however the study shows some weaknesses in the approach and conclusions as highlighted in the comments below:
1- The rationale behind initiating the study with leukemia cells HAP-1 is unclear.
2- Fig.1C-D, it would be more informative to show the growth of WT and KO cells in the presence of MgCl2
3- Fig.1E: is the of NS8593 effect only observed at 72h post treatment? And is the decrease in viability all due to cell death or does the compound also elicit anti-proliferative effects?
4- Fig. 1B and Fig 1E, could the authors explain why NS8593 blocks close to 80% of the current at 10 µM but doesn’t affect viability? Are the effects observed in Fig.1E due to TRPM7 inhibition or do the authors suspect some off-target effect of the compound?
5- Fig.2 I applaud the authors for using the Venn diagram to illustrate the overlap between two TRPM7-KO clones and NS8593-treated cells to determine targets of TRPM7. Could the authors clarify why they pursued HER2 as a target even though the pharmacological inhibition of TRPM7 did not show the same result as the KO. ALPK1 however was upregulated in both KO clones and NS8593-treated cells, yet the authors haven’t pursued this target.
6- Fig.3 Could the authors compare the relative changes in HER2 expression normalized to controls? It appears from the results shown that Zn and ZP have similar effect on HER2 in WT and KO cells, which may indicate an independent effect of Zinc on HER2 expression that is not related to TRPM7.
7- Have the authors considered investigating magnesium effect on HER2 expression, given the role of TRPM7 in magnesium homeostasis?
8- Fig.4 the data from this figure show contradictory effect of TRPM7 KO in other cells (TS and HEK293) than observed in HAP1 cells. Does this mean the effects in HAP-1 cells are cell line specific Or unrelated to TRPM7’s true downstream targets?
9- Fig.5 if the the authors hypothesis is that TRPM7 regulates HER2 expression, could they discuss the absence of correlation between TRPM7 expression and HER2 expression as well as their choice of cell lines. Eg. SKBR3 and MDA-MB231 have similar current size, yet one is HER2 high and the other is HER2 deficient. Due to the anomaly in MDA-MB231, being triple negative, this cell line may not be the appropriate control to investigate the role of TRPM7 in regulating HER2.
10- Fig.6 the use of NS8593 seems inappropriate in studying the synergistic effect of TRPM7 and HER2. According to the authors’ data, NS8593 has not shown any effect on HER2 and all the changes in HER2 expression (upregulation or downregulation) were shown using KO or siRNA. It would be more suitable to use a KO or siRNA in this experiment.
11- Overall, the data from this study showed contradicting effects of TRPM7 KO on HER2 depending on which cell line is used, which makes it difficult to conclude a connection between the two genes.
12- HER2 breast cancer generally have good prognosis and effective therapeutic options. Could the authors discuss how TRPM7 would be any beneficial to existing targeted therapies against HER2 positive breast cancer?
Reviewer 4 Report
Comments and Suggestions for Authors
The paper is well-structured, with a logical flow from introduction to conclusions. The research is grounded in solid scientific methods, and the results are discussed in the context of their potential to lead to new cancer treatments. The study is a significant contribution to the field, highlighting a novel aspect of cancer biology that could lead to more effective therapies for patients with HER2-positive breast cancer and potentially other cancers. The combination of genetic and pharmacological approaches enriches the study's impact, making a compelling case for further exploration of TRPM7 as a therapeutic target. Ithink the paper should be accepted after minor reversion.
Minor:
The abstract efficiently introduces the significance of TRPM7 in cancer progression and its dual functionality as a channel and kinase. It outlines the research objective to analyze the impact of NS8593, a TRPM7 inhibitor, and TRPM7 gene knockout on leukemia cells and potentially on breast cancer treatment. This summary sets a strong foundation for understanding the study’s scope and its implications for cancer therapy.
The introduction is well-crafted, highlighting the biochemical properties and physiological roles of TRPM7, particularly in divalent cation transport critical to cellular function and cancer pathology. The narrative builds a compelling case for TRPM7 as a therapeutic target, backed by previous studies that underscore its role in cancer cell proliferation and the modulation of various cancer-related pathways. However, some previous studies was not mentioned, especially works from one of the most famous groups in the TRPM7 field, Jun Lin’s lab from Stony Brook US, should definitely be emphasized. One paper recommended is PMID: 36332746, this reviewed the general information of TRPM7 in different cancers. Also, their study PMID: 31947967 and PMID: 33435261, as I know is the first study using TRPM7 knockout MDA-MB-231. The fundamental contribution of this group for TRPM7 cancer study should be emphasized and this reference definitely should be mentioned in the paper. Another group should be mentioned is the Frank N. van Leeuwen’s lab and their study in TRPM7 cancer metastasis, PMID: 22871386. In addition, Halima Ouadid-Ahidouch’s lab from Univ. Picardie Jules Verne contributed to the understanding of TRPM7 in breast cancers. This work definitely as an example PMID: 19515901, on the other hand, in another of their study, the role of TRPM7 kinase domain was studied, PMID: 23910495. These references should be cited (which is essential) to improve the reader's understanding of a few minestrone in TRPM7 cancer research.
Materials and Methods section meticulously details the methodologies employed, from cell line preparation and genetic manipulation to pharmacological treatments and electrophysiological measurements. The use of both genome editing and pharmacological inhibition provides a robust approach to elucidate TRPM7's role in cancer. Methods like Western blotting, patch-clamp techniques, and cell viability assays are appropriately described, ensuring the study's experimental rigor. This section is complete and great. Great job!
The results are presented with clarity and depth, demonstrating how TRPM7 knockout and inhibition affect cancer cell proliferation and gene expression. Significant findings include the upregulation of HER2 expression in TRPM7-deficient cells and its modulation through zinc supplementation, suggesting a novel regulatory mechanism via TRPM7. The electrophysiological data corroborate the genetic data, providing a comprehensive view of TRPM7's impact at the molecular and cellular levels. This particularly interesting. I enjoyed reading it.
The discussion integrates the study's findings with the current understanding of TRPM7’s role in cancer, emphasizing the potential for dual inhibition strategies targeting TRPM7 and HER2 in breast cancer therapy. The discussion thoughtfully considers the implications of TRPM7's zinc regulation on HER2 expression, offering insightful hypotheses about the interaction between these two molecules in various cancer cell models. The exploration of TRPM7’s broader role across different cell types and the consistency of its impact on HER2 across these models are particularly well-articulated. The paper concludes with a strong summary of the findings and their implications for cancer therapy, particularly in HER2-positive breast cancer. It highlights the need for further research to explore TRPM7’s interactions with other divalent cations and their roles in cancer biology. The suggestion of combinatorial therapies provides a promising avenue for future clinical applications.However, Similar issue as the introduction, the authors does not familiar with the major research in the field and did not provide enough comparison of their results with previous studies. This must improve before the acceptance of the paper. Specifically, the above-mentioned papers as well as other similar studies should be discussed, for both method comparing and result comparing. In addition, the authors should also expand the TRPM7 and their studies to some critical fields of breast cancer treatment. Such as 1) Pharmacological hormone therapies in breast cancer, this recent review PMID: 38909530 introduced the history of breast cancer hormone therapies in one of the section, 2) HER2 associated drug resistance issue PMID: 38503142. 3) Non-invasive detection of breast cancer PMID: 38134587, these references are highly recommended, but more other citation is needed as well.
Round 2
Reviewer 3 Report
Comments and Suggestions for Authors
No further comments.